# Deep Random Splines for Point Process Intensity Estimation

**Gabriel Loaiza-Ganem & John P. Cunningham**
Department of Statistics
Columbia University
New York City, NY 10027, USA
`{gl2480, jpc2181}@columbia.edu`

## Abstract

Gaussian processes are the leading class of distributions on random functions, but they suffer from well known issues including difficulty scaling and inflexibility with respect to certain shape constraints (such as nonnegativity). Here we propose Deep Random Splines, a flexible class of random functions obtained by transforming Gaussian noise through a deep neural network whose output are the parameters of a spline. Unlike Gaussian processes, Deep Random Splines allow us to readily enforce shape constraints while inheriting the richness and tractability of deep generative models. We also present an observational model for point process data which uses Deep Random Splines to model the intensity function of each point process and apply it to neuroscience data to obtain a low-dimensional representation of spiking activity. Inference is performed via a variational autoencoder that uses a novel recurrent encoder architecture that can handle multiple point processes as input.

## 1 Introduction

Gaussian Processes (GPs) are one of the main tools for modeling random functions (Rasmussen, 2004). They allow control of the smoothness of the function by choosing an appropriate kernel but have the disadvantage that, except in special cases (for example Gilboa et al. (2015); Flaxman et al. (2015)), inference in GP models scales poorly in both memory and runtime. Furthermore, GPs cannot easily handle shape constraints. It can often be of interest to model a function under some shape constraint, for example nonnegativity, monotonicity or convexity/concavity (Møller et al., 1998; Schmidt & Hess, 1988; Ramsay, 1988; Mammen, 1991). While some shape constraints can be enforced by transforming the GP or by enforcing them at a finite number of points, doing so cannot always be done and usually makes inference harder, see for example Lin & Dunson (2014).

Splines are another popular tool for modeling unknown functions (Wahba, 1990). When there are no shape constraints, frequentist inference is straightforward and can be performed using linear regression, by writing the spline as a linear combination of basis functions. Under shape constraints, the basis function expansion usually no longer applies, since the space of shape constrained splines is not typically a vector space. However, the problem can usually still be written down as a tractable constrained optimization problem (Schmidt & Hess, 1988). Furthermore, when using splines to model a random function, a distribution must be placed on the spline's parameters, so the inference problem becomes Bayesian. DiMatteo et al. (2001) proposed a method to perform Bayesian inference in a setting without shape constraints, but the method relies on the basis function expansion and cannot be used in a shape constrained setting. Furthermore, fairly simple distributions have to be placed on the spline parameters for their approximate posterior sampling algorithm to work adequately, which results in the splines having a restrictive and oversimplified distribution.

On the other hand, deep probabilistic models take advantage of the major progress in neural networks to fit rich, complex distributions to data in a tractable way (Rezende et al., 2014; Mohamed & Lakshminarayanan, 2016; Kingma & Welling, 2013; Gao et al., 2016; Johnson et al., 2016). However, their goal is not usually to model random functions.

In this paper, we introduce Deep Random Splines (DRS), an alternative to GPs for modeling random functions. DRS are a deep probabilistic model in which standard Gaussian noise is transformed through a neural network to obtain the parameters of a spline, and the random function is then the corresponding spline. This combines the complexity of deep generative models and the ability to enforce shape constraints of splines.

We use DRS to model the nonnegative intensity functions of Poisson processes (Kingman, 1992). In order to ensure that the splines are nonnegative, we use a parametrization of nonnegative splines that can be written as an intersection of convex sets, and then use the method of alternating projections (von Neumann, 1950) to obtain a point in that intersection (and differentiate through that during learning). To perform scalable inference, we use a variational autoencoder (Kingma & Welling, 2013) with a novel encoder architecture that takes multiple, truly continuous point processes as input (not discretized in bins, as is common).

Our contributions are: $(i)$ Introducing DRS, $(ii)$ using the method of alternating projections to constrain splines, $(iii)$ proposing a variational autoencoder model whith a novel encoder architecture for point process data which uses DRS, and $(iv)$ showing that our model outperforms commonly used alternatives in both simulated and real data.

The rest of the paper is organized as follows: we first explain DRS, how to parametrize them and how constraints can be enforced in section 2. We then present our model and how to do inference in section 3. We then compare our model against competing alternatives in simulated data and in two real spiking activity datasets in section 4, and observe that our method outperforms the alternatives. Finally, we summarize our work in section 5.

## 2 DEEP RANDOM SPLINES

Throughout the paper we will consider functions on the interval $[T_1, T_2)$ and will select $I + 1$ fixed knots $T_1 = t_0 < \cdots < t_I = T_2$. We will refer to a function as a spline of degree $d$ and smoothness $s < d$ if the function is a $d$-degree polynomial in each interval $[t_i, t_{i+1})$ for $i = 0, \ldots, I - 1$, is continuous, and $s$ times differentiable. We will denote the set of splines of degree $d$ and smoothness $s$ by $\mathcal{G}_{d,s} = \{g_\psi : \psi \in \Psi_{d,s}\}$, where $\Psi_{d,s}$ is the set of parameters of each polynomial in each interval. That is, every $\psi \in \Psi_{d,s}$ contains the parameters of each of the $I$ polynomial pieces (it does not contain the locations of the knots as we take them to be fixed since we observed overfitting when not doing so). While the most natural ways to parametrize splines of degree $d$ are a linear combination of basis functions or with the $d + 1$ polynomial coefficients of each interval, these parametrizations do not lend themselves to easily enforce constraints such as nonnegativity (Schmidt & Hess, 1988). We will thus use a different parametrization which we will explain in detail in the next section. We will denote by $\Psi \subseteq \Psi_{d,s}$ the subset of spline parameters that result in the splines having the shape constraint of interest, for example, nonnegativity.

DRS are a distribution over $\mathcal{G}_{d,s}$. To sample from a DRS, a standard Gaussian random variable $Z \in \mathbb{R}^m$ is transformed through a neural network parametrized by $\theta$, $f_\theta : \mathbb{R}^m \to \Psi$. The DRS is then given by $g_{f_\theta(Z)}$ and inference on $\theta$ can be performed through a variational autoencoder (Kingma & Welling, 2013). Note that $f$ maps to $\Psi$, thus ensuring that the spline has the relevant shape constraint.

### 2.1 CONSTRAINING SPLINES

We now explain how we can enforce piecewise polynomials to form a nonnegative spline. We add the nonnegativity constraint to the spline as we will use it for our model in section 3, but constraints such as monotonicity and convexity/concavity can be enforced in an analogous way. In order to achieve this, we use a parametrization of nonnegative splines that might seem overly complicated at first. However, it has the critical advantage that it decomposes into the intersection of convex sets that are easily characterized in terms of the parameters, which is not the case for the naive parametrization which only includes the $d + 1$ coefficients of every polynomial. We will see how to take advantage of this fact in the next section.

A beautiful but perhaps lesser known spline result (see Lasserre (2010)) gives that a polynomial $p(t)$ of degree $d$, where $d = 2k + 1$ for some $k \in \mathbb{N}$, is nonnegative in the interval $[l, u)$ if and only if it

can be written down as follows:

$$p(t) = (u - t)[t]^\top Q_1[t] + (t - l)[t]^\top Q_2[t] \tag{1}$$

where $[t] = (1, t, t^2, \ldots, t^k)^\top$ and $Q_1$ and $Q_2$ are $(k+1) \times (k+1)$ symmetric positive semidefinite matrices. It follows that a piecewise polynomial of degree $d$ with knots $t_0, \ldots, t_I$ defined as $p^{(i)}(t)$ for $t \in [t_{i-1}, t_i)$ for $i = 1, \ldots, I$ is nonnegative if and only if it can be written as:

$$p^{(i)}(t) = (t_i - t)[t]^\top Q_1^{(i)}[t] + (t - t_{i-1})[t]^\top Q_2^{(i)}[t] \tag{2}$$

for $i = 1, \ldots, I$, where each $Q_1^{(i)}$ and $Q_2^{(i)}$ are $(k+1) \times (k+1)$ symmetric positive semidefinite matrices. We can thus parametrize every piecewise nonnegative polynomial on our $I$ intervals with $(Q_1^{(i)}, Q_2^{(i)})_{i=1}^I$. If no constraints are added on these parameters, the resulting piecewise polynomial might not be smooth, so certain constraints have to be added in order to guarantee that we are parametrizing a nonnegative spline and not just a nonnegative piecewise polynomial. To that end, we define $\mathcal{C}_1$ as the set of $(Q_1^{(i)}, Q_2^{(i)})_{i=1}^I$ such that:

$$p^{(i)}(t_i) = p^{(i+1)}(t_i) \text{ for } i = 1, \ldots, I-1 \tag{3}$$

That is, $\mathcal{C}_1$ is the set of parameters whose resulting piecewise polynomial as in equation 2 is continuous. Analogously, let $\mathcal{C}_j$ for $j = 2, 3, \ldots$ be the set of $(Q_1^{(i)}, Q_2^{(i)})_{i=1}^I$ such that:

$$\frac{\partial^{j-1}}{\partial t^{j-1}} p^{(i)}(t_i) = \frac{\partial^{j-1}}{\partial t^{j-1}} p^{(i+1)}(t_i) \text{ for } i = 1, \ldots, I-1 \tag{4}$$

So that $\mathcal{C}_j$ is the set of parameters whose corresponding piecewise polynomials have matching left and right $(j-1)$-th derivatives. Let $\mathcal{C}_0$ be the set of $(Q_1^{(i)}, Q_2^{(i)})_{i=1}^I$ which are symmetric positive semidefinite. We can then parametrize the set of nonnegative splines on $[T_1, T_2)$ by $\Psi = \cap_{j=0}^{s+1} \mathcal{C}_j$. Note that the case where $d$ is even can be treated analogously (see supplementary material).

## 2.2 THE METHOD OF ALTERNATING PROJECTIONS

In order to use a DRS, $f_\theta$ has to map to $\Psi$, that is, we need to have a way for a neural network to map to the parameter set corresponding to nonnegative splines. We achieve this by taking $f_\theta(z) = h(\tilde{f}_\theta(z))$, where $\tilde{f}_\theta$ is an arbitrary neural network and $h$ is a surjective function onto $\Psi$. The most natural choice for $h$ is the projection onto $\Psi$. However, while computing the projection onto $\Psi$ (for $\Psi$ as in section 2.1) can be done by solving a convex optimization problem, it cannot be done analytically. This is an issue because when we train the model, we will need to differentiate $f_\theta$ with respect to $\theta$. Note that Amos & Kolter (2017) propose a method to have an optimization problem as a layer in a neural network. One might hope to use their method for our problem, but it cannot be applied due to the semidefinite constraint on our matrices.

The method of alternating projections (von Neumann, 1950; Bauschke & Borwein, 1996) allows us to approximately compute such a function $h$ analytically. If $\mathcal{C}_0, \ldots, \mathcal{C}_{s+1}$ are closed, convex sets in $\mathbb{R}^D$, then the sequence $\psi^{(k)} = P_{k \bmod (s+2)}(\psi^{(k-1)})$ converges to a point in $\cap_{j=0}^{s+1} \mathcal{C}_j$ for any starting $\psi^{(0)}$, where $P_j$ is the projection onto $\mathcal{C}_j$ for $j = 0, \ldots, s+1$. The method of alternating projections then consists on iteratively projecting onto each set in a cyclic fashion. We call computing $\psi^{(k)}$ from $\psi^{(k-1)}$ the $k$-th iteration of the method of alternating projections. This method can be useful to obtain a point in the intersection if each $P_j$ can be easily computed.

In our case, projecting onto $\mathcal{C}_0$ can be done by doing eigenvalue decompositions of $Q_1^{(i)}$ and $Q_2^{(i)}$ and zeroing out negative elements in the diagonal matrices containing the eigenvalues. While this might seem computationally expensive, the matrices are small and this can be done efficiently. For example, for cubic splines ($d = 3$), there are $2I$ matrices each one of size $2 \times 2$. Projecting onto $\mathcal{C}_j$ for $j = 1, \ldots s+1$ can be done analytically as it can be formulated as a quadratic optimization problem with linear constraints. Furthermore, because of the local nature of the constraints where every interval is only constrained by its neighboring intervals, this quadratic optimization problem can be reduced to solving a tridiagonal system of linear equations of size $I - 1$ which can be solved efficiently in $O(I)$ time with simplified Gaussian elimination. While the derivation of this fact is a

straightforward application of the KKT conditions, the algebra is cumbersome, so we omit it here to include it in the supplementary material.

By letting $h$ be the first $M$ iterations of the method of alternating projections, we can ensure that $f_\theta$ maps (approximately) to $\Psi$, while still being able to compute $\nabla_\theta f_\theta(z)$. Note that we could find such an $h$ function using Dykstra's algorithm (not to be confused with Dijkstra's shortest path algorithm), which is a modification of the method of alternating projections that converges to the projection of $\psi^{(0)}$ onto $\cap_{j=0}^{s+1} C_j$ (Dykstra, 1983; Boyle & Dykstra, 1986; Tibshirani, 2017)), but we found that the method of alternating projections was faster to differentiate when using reverse mode automatic differentiation packages (Abadi et al., 2016).

Another way of finding such an $h$ would be unrolling any iterative optimization method that solves the projection onto $\Psi$, such as gradient-based methods or Newton methods. We found the alternating projections method more convenient as it does not involve additional hyperparameters such as learning rate that drastically affect performance. Furthermore, the method of alternating projections is known to have a linear convergence rate (as fast as gradient-based methods) that is independent of the starting point (Bauschke & Borwein, 1996). This last observation is important, as the starting point in our case is determined by the output of $\tilde{f}_\theta$, so that the convergence rate being independent of the starting point ensures that $\tilde{f}_\theta$ cannot learn to ignore $h$, which is not the case for gradient-based and Newton methods (for a fixed number of iterations and learning rate, there might exist an initial point that is too far away to actually reach the projection). Finally, note that if we wanted to enforce, for example, that the spline be monotonic, we could parametrize its derivative and force it to be nonnegative or nonpositive. Convexity or concavity can be enforced analogously.

## 3 Deep Random Splines as Intensity Functions of Point Processes

Since we will use DRS as intensity functions for Poisson processes, we begin this section with a brief review of these processes.

### 3.1 Poisson Processes

An inhomogeneous Poisson process in a set $\mathcal{S}$ is a random subset of $\mathcal{S}$. The process can (for our purposes) be parametrized by an intensity function $g : \mathcal{S} \to \mathbb{R}_+$ and in our case, $\mathcal{S} = [T_1, T_2)$. We write $S \sim \mathcal{PP}_\mathcal{S}(g)$ to denote that the random set $S$, whose elements we call events, follows a Poisson process on $\mathcal{S}$ with intensity $g$. If $S = \{x_k\}_{k=1}^K \sim \mathcal{PP}_\mathcal{S}(g)$, then $|S \cap A|$, the number of events in any $A \subseteq \mathcal{S}$, follows a Poisson distribution with parameter $\int_A g(t)dt$ and the log likelihood of $S$ is given by:

$$\log p(\{x_k\}_{k=1}^K | g) = \sum_{k=1}^K \log g(x_k) - \int_\mathcal{S} g(t)dt \qquad (5)$$

Splines have the very important property that they can be analytically integrated (as the integral of polynomials can be computed in closed form), which allows to exactly evaluate the log likelihood in equation 5 when $g$ is a spline. As a consequence, fitting a DRS to observed events is more tractable than fitting models that use GPs to represent $g$, such as log-Gaussian Cox processes (Møller et al., 1998). Inference in the latter type of models is very challenging, despite some efforts by Cunningham et al. (2008); Adams et al. (2009); Lloyd et al. (2015). Splines also vary smoothly, which incorporates the reasonable assumption that the expected number of events changes smoothly over time. These properties were our main motivations for choosing splines to model intensity functions.

### 3.2 Our Model

Suppose we observe $N$ simultaneous point processes in $[T_1, T_2)$ a total of $R$ repetitions (we will call each one of these repetitions/samples a trial). Let $X_{r,n}$ denote the $n$-th point process of the $r$-th trial. Looking ahead to an application we study in the results, data of this type is a standard setup for microelectrode array data, where $N$ neurons are measured from time $T_1$ to time $T_2$ for $R$ repetitions, and each event in the point processes corresponds to a spike (the time at which the neurons "fired"). Each $X_{r,n}$ is also called a spike train. The model we propose, which we call DRS-VAE, is as follows:

$$\begin{cases} Z_r \sim \mathcal{N}(0, I_m) \text{ for } r = 1, \ldots, R \\ \psi_{r,n} = f_\theta^{(n)}(Z_r) \text{ for } n = 1, \ldots, N \\ X_{r,n} | \psi_{r,n} \sim \mathcal{PP}_{[T_1, T_2]}(g_{\psi_{r,n}}) \end{cases} \tag{6}$$

where each $f_\theta^{(n)} : \mathbb{R}^m \to \Psi$ is obtained as described in section 2.2. The hidden state $Z_r$ for the $r$-th trial $\mathbf{X}_r := (X_{r,1}, \ldots, X_{r,N})$ can be thought as a low-dimensional representation of $\mathbf{X}_r$. Note that while the intensity function of every point process and every trial is a DRS, the latent state $Z_r$ of each trial is shared among the $N$ point processes.

Once again, one might think that our parametrization of nonnegative splines is unnecessarily complicated and that having $f_\theta^{(n)}$ in equation 6 be a simpler parametrization of an arbitrary spline (e.g. basis coefficients) and using $\tau(g_{\psi_{r,n}})$ instead of $g_{\psi_{r,n}}$, where $\tau$ is a nonnegative function, might be a better solution to enforcing nonnegativity constraints. The function $\tau$ would have to be chosen in such a way that the integral of equation 5 can still be computed analytically, making $\tau(t) = t^2$ a natural choice. While this would avoid having to use the method of alternating projections, we found that squared splines perform very poorly as they oscillate too much.

### 3.3 INFERENCE

Autoencoding variational Bayes (Kingma & Welling, 2013) is a technique to perform inference in the following type of model:

$$\begin{cases} Z_r \sim p_\theta(z) \text{ for } r = 1, \ldots, R \\ X_r \sim p_\theta(x|z_r) \end{cases} \tag{7}$$

where each $Z_r \in \mathbb{R}^m$ is a local hidden variable which we do not observe, $\theta$ are the model parameters and $X_r$ is the data that we actually observe, whose distribution depends on $Z_r$. A variational autoencoder estimates $\theta$ and approximates the posterior $p(\mathbf{z}|\mathbf{x})$ by a distribution $q_\phi(\mathbf{z}|\mathbf{x})$ parametrized by $\phi$. Further simplifying assumptions are made and $q_\phi(\mathbf{z}|\mathbf{x})$ is taken such that it respects conditional independence:

$$q_\phi(\mathbf{z}|\mathbf{x}) = \prod_{r=1}^{R} q_\phi(z_r|x_r) \tag{8}$$

where each $q_\phi(z_r|x_r)$ is taken to be normal with mean and variance depending on $\mathbb{R}^m$ valued nonlinear functions (usually taken to be neural networks) of $x_r$:

$$q_\phi(z_r|x_r) = \mathcal{N}\Big(\mu_\phi(x_r), \operatorname{diag}\big(\sigma_\phi^2(x_r)\big)\Big) \tag{9}$$

where $\operatorname{diag}\big(\sigma_\phi^2(x_r)\big)$ is a diagonal matrix whose diagonal elements are given by $\sigma_\phi^2(x_r)$. Performing (approximate) Bayesian inference becomes finding values of $\phi$ that adequately approximate the true posterior. To achieve this task, the ELBO $\mathcal{L}$, which is given by the following expression, is jointly maximized over $(\theta, \phi)$:

$$\mathcal{L}(\theta, \phi) = \sum_{r=1}^{R} -KL(q_\phi(z_r|x_r)||p_\theta(z_r)) + E_{q_\phi(z_r|x_r)}[\log p_\theta(x_r|z_r)] \tag{10}$$

Maximizing the ELBO over $\phi$ is equivalent to minimizing the KL of the approximate posterior to the true posterior (for a fixed $\theta$), while maximizing it over $\theta$ is equivalent to maximizing a lower bound on the log likelihood. This lower bound is close to the actual log likelihood when the true posterior is correctly approximated. Furthermore, the first term in the sum in equation 10 can be written down in closed form as it is just the KL divergence between two normal random variables, while the second term in the sum can be written using the reparametrization trick:

$$E_{q_\phi(z_r|x_r)}[\log p_\theta(x_r|z_r)] = E_{\epsilon \sim \mathcal{N}(0, I_m)}[\log p_\theta(x_r|\mu_\phi(x_r) + \sigma_\phi(x_r) \odot \epsilon)] \tag{11}$$

where $\odot$ refers to coordinate-wise multiplication. This allows for straightforward differentiation with respect to $\phi$, and thus stochastic gradient methods can be used.

In order to perform inference, we use autoencoding variational Bayes. Because of the point process nature of the data, $\mu_\phi$ and $\sigma_\phi$ require a recurrent architecture, since their input $\mathbf{x}_r = (x_{r,1}, x_{r,2}, \ldots, x_{r,N})$ consists of $N$ point processes. This is challenging because the input is not just a sequence, but $N$ sequences of different lengths (numbers of events). In order to deal with this, we use N separate LSTMs (Hochreiter & Schmidhuber, 1997), one per point process. Each LSTM takes as input the events of the corresponding point process. The final states of each LSTM are then concatenated and transformed through a dense layer (followed by an exponential activation in the case of $\sigma_\phi$ to ensure positivity) in order to map to the hidden space $\mathbb{R}^m$. We also tried bidirectional LSTMs (Graves & Schmidhuber, 2005) but found regular LSTMs to be faster while having similar performance. The architecture is depicted in figure 1. Combining equations 10 and 11 for our model of equation 6, we approximate the ELBO at each stochastic gradient step by:

$$
\begin{aligned}
\mathcal{L}(\theta, \phi) \approx {} & \frac{R}{|\mathcal{B}|} \sum_{r \in \mathcal{B}} -KL(q_\phi(z_r | \mathbf{x}_r) || p_\theta(z_r)) \\
& + \frac{1}{L} \sum_{l=1}^{L} \log p_\theta(x_r | \mu_\phi(\mathbf{x}_r) + \sigma_\phi(\mathbf{x}_r) \odot \epsilon_l) \\
= {} & \frac{R}{|\mathcal{B}|} \sum_{r \in \mathcal{B}} \frac{1}{2} \sum_{j=1}^{m} \left( 1 + \log \sigma_{\phi,j}^2(\mathbf{x}_r) \right. \\
& \qquad\qquad\qquad\qquad \left. - \mu_{\phi,j}(\mathbf{x}_r)^2 - \sigma_{\phi,j}^2(\mathbf{x}_r) \right) \\
& + \frac{R}{|\mathcal{B}|} \sum_{r \in \mathcal{B}} \frac{1}{L} \sum_{l=1}^{L} \sum_{n=1}^{N} \sum_{k=1}^{K_{r,n}} \log g_{\psi_{r,n,l}}(x_{r,n,k}) \\
& - \int_{T_1}^{T_2} g_{\psi_{r,n,l}}(t) dt
\end{aligned}
\tag{12}
$$

where $\mathcal{B}$ is a randomly selected subset of trials, $\epsilon_1, \ldots \epsilon_L$ are iid $\mathcal{N}(0, I_m)$, $\mu_{\phi,j}(\mathbf{x}_r)$ and $\sigma_{\phi,j}^2(\mathbf{x}_r)$ are, respectively, the $j$-th coordinates of $\mu_\phi(\mathbf{x}_r)$ and $\sigma_\phi^2(\mathbf{x}_r)$, $K_{r,n}$ is the number of events in the $n$-th point process of the $r$-th trial, $\psi_{r,n,l} = f_\theta^{(n)}(\mu_\phi(\mathbf{x}_r) + \sigma_\phi(\mathbf{x}_r) \odot \epsilon_l)$ and $x_{r,n,k}$ is the $k$-th event of the $n$-th point process of the $r$-th trial.

Gao et al. (2016) have a similar model, where a hidden Markov model is transformed through a neural network to obtain event counts on time bins. The hidden state for a trial in their model is then an entire hidden Markov chain, which will have significantly higher dimension than our hidden state. Also, their model can be recovered from ours if we change the standard Gaussian distribution of $Z_r$ in equation 6 to reflect their Markovian structure and choose $\mathcal{G}$ to be piecewise constant, nonnegative functions. We also emphasize the fact that our model is very easy to extend: for example, it would be straightforward to extend it to multi-dimensional point processes (not neural data any more) by changing $\mathcal{G}$ and its parametrization. It is also straightforward to use a more complicated point process than the Poisson one by allowing the intensity to depend on previous event history. Furthermore, DRS can be used in settings that require random functions, even if no point process is involved.

## 4 Experiments

### 4.1 Simulated Data

We simulated data with the following procedure: First, we set 2 different types of trials. For each type of trial, we sampled one true intensity function on $[0, 10]$ for each of the $N = 2$ point processes by sampling from a GP and exponentiating the result. We then sampled 600 times from each type of trial, resulting in 1200 trials. We randomly selected 1000 trials for training and set aside the rest for testing. We then fit the model described in section 3.2 and compared it against the PfLDS model of Gao et al. (2016) and the GPFA model of Yu et al. (2009). Both of these methods discretize time into $B$ time bins and have a latent variable per time bin and per trial (as opposed to our model which is only per trial). They do this as a way of enforcing temporal smoothness by placing an appropriate prior over their latent trajectories, which we do not have to do as we implicitly enforce temporal smoothness by using splines to model intensity functions. PfLDS uses Gaussian linear dynamics

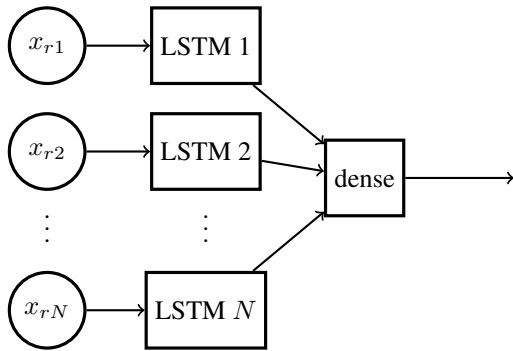

Figure 1: Each point process $x_{r1}, \ldots, x_{rN}$ is processed by its corresponding LSTM and everything is joined by a fully-connected layer at the end.

for their latent space and a Poisson distribution on the number of events per time bin, while GPFA places a GP distribution on the latent space and a Gaussian distribution on the square-rooted number of events per time bin. We compare these methods against DRS-VAE since both were designed to analyze the same type of point process data and, inference wise, PfLDS uses a similar autoencoding variational Bayes algorithm whereas GFPA uses GPs to model random functions.

We used a uniform grid with 11 knots (resulting in $I = 10$ intervals), picked $d = 3$, $s = 2$, used a mini-batch size of 2 and used $L = 2$. The state of each LSTM has 100 units, and $\tilde{f}$ is a feed-forward neural network with ReLU activations and with 3 hidden layers, each one with 100 units. We apply 102 iterations of the method of alternating projections. Since a twice-differentiable cubic spline on $I$ intervals has $I + 3$ degrees of freedom, when discretizing time for PfLDS and GPFA we use $B = I + 3 = 13$ time bins. This way the distribution recovered by PfLDS also has $B = 13$ degrees of freedom, while the distribution recovered by GPFA has even more, as each Gaussian has a covariance in addition to the mean. We set the latent dimension $m$ in our model to 2 and we also set the latent dimension per time bin in PfLDS and GPFA to 2, meaning that the overall latent dimension for an entire trial was $2B = 26$. These two choices make the comparison conservative as they allow more flexibility for the two competing methods than for ours. For the feed-forward architecture in PfLDS, we also used 3 hidden layers, each with 100 units.

The left panel of figure 2 shows the posterior means of the hidden variables in our model for each of the 200 test trials. Each posterior mean is colored according to its type of trial. We can see that different types of trials form separate clusters, meaning that our model successfully obtains low-dimensional representations of the trials. Note that the model is trained without having access to the type of each trial; colors are assigned in the figure post hoc.

The right panel of figure 2 shows the events (in red) for a particular point process on a particular trial, along with the true intensity (in green) that generated the events and posterior samples from our model (in purple) and from PfLDS (in blue) of the corresponding intensities. Note that since PfLDS parametrizes the number of counts on each time bin, it does not have a corresponding intensity. We plot instead a piecewise constant intensity on each time bin in such a way that the resulting Poisson process has the same count distribution as the distribution that is parametrized by PfLDS. We can see that our method recovers a smoother function that is closer to the truth than PfLDS.

Table 1: Quantitative comparison of our method (DRS-VAE) against PfLDS and GPFA on simulated data.

| METHOD | ELBO | $L^2$ | p-VALUE |
|---|---|---|---|
| DRS-VAE | **57.17** | **$0.11 \pm 0.086$** | $-$ |
| PfLDS | 52.32 | $0.21 \pm 0.103$ | $6.6 \times 10^{-45}$ |
| GPFA | $-$ | $0.21 \pm 0.097$ | $4.7 \times 10^{-46}$ |

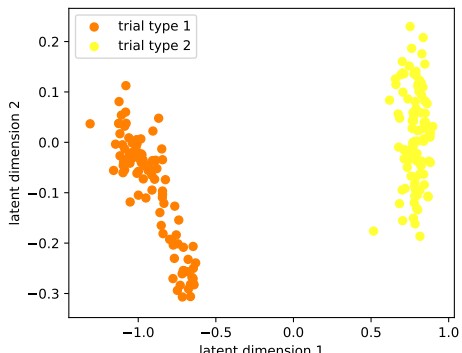 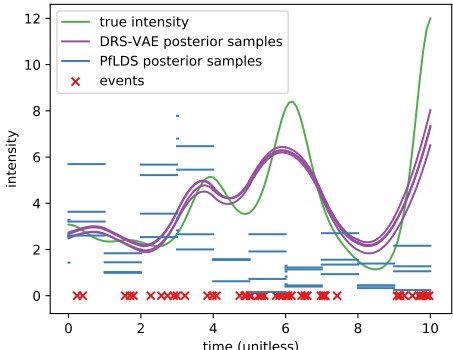

Figure 2: Posterior means of the hidden variables of DRS by type of trial on simulated data (left panel) and comparison of posterior intensities of our method (DRS-VAE) against PfLDS (right panel).

Table 1 shows performance from our model compared against PfLDS and GPFA. The second column shows the per-trial ELBO on test data, and we can see that our model has a larger ELBO than PfLDS. While having a better ELBO does not imply that our log likelihood is better, it does suggest that it is. Since both PfLDS and GPFA put a distribution on event counts on time bins instead of a distribution on event times as our models does, the log likelihoods are not directly comparable. However, in the case of PfLDS, we can easily convert from the Poisson likelihood on time bins to the piecewise constant intensity Poisson process likelihood, so that the numbers become comparable. In order to get a quantitative comparison between our model and GPFA, we take advantage of the fact that we know the true intensity that generated the data and compare average $L^2$ distance, across point processes and trials, between posterior intensity samples and actual intensity function. Once again, we can see that our method outperforms the alternatives. Table 1 also includes the standard deviation of these $L^2$ distances. Since the standard deviations are somewhat large in comparison to the means, for each of the two competing alternatives, we carry out a two sample t-test comparing the $L^2$ distance means obtained with our method against the alternative. The $p$-values indicate that our method recovers intensity functions that are closer to the truth in a statistically significant way.

## 4.2 REAL DATA

### 4.2.1 REACHING DATA

We also fit our model to the dataset collected by Churchland et al. (2012). The dataset, after preprocessing (see supplementary material for details), consists of measurements of 20 neurons for 3590 trials on the interval $[-100, 300)$ (in $ms$) of a primate. In each trial, the primate reaches with its arm to a specific location, which changes from trial to trial (we can think of the 40 locations as types of trials), where time 0 corresponds to the beginning of the movement. We randomly split the data into a training set with 3000 trials and a test set with the rest of the trials.

We chose $d = 3$, $s = 2$, $L = 2$, applied 102 iterations of the method of alternating projections, set the state of each LSTM to have 25 units and $\tilde{f}$ is a feed-forward network with ReLU activations and with 3 hidden layers, each one with 10 units (we tried more complicated architectures but saw no improvement). We used 18 uniformly spaced knots (that is, 17 intervals). For the comparison against PfLDS, we split time into 20 bins, resulting in time bins of $20ms$ (which is a standard length), once again making sure that the degrees of freedom are comparable. Since we do not have access to the ground truth, we do not compare against GPFA as the $L^2$ metric computed in the previous section cannot be used here. Again, we used a hidden dimension $m = 2$ for our model, resulting in hidden trajectories of dimension 40 for PfLDS. We experimented with larger values of $m$ but did not observe significant improvements in either model.

Figure 3 shows the spike train (red) for a particular neuron on a particular trial, along with posterior samples from our model (in purple) and from PfLDS (in blue) of the corresponding intensities. We can see that the posterior samples look like plausible intensities to have generated the corresponding spike trains and that our posterior intensities look smoother than the ones obtained by PfLDS.

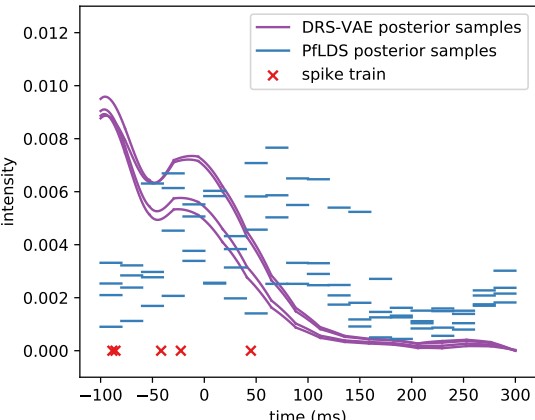

Figure 3: Comparison of posterior intensities of our method (DRS-VAE) against PfLDS on reaching data.

Table 2: Quantitative comparison of our method (DRS-VAE) against PfLDS on reaching data.

| MODEL | ELBO | 15-NN | SSG/SST |
|---|---|---|---|
| DRS-VAE | $-\mathbf{500.77}$ | $\mathbf{23.73}\%$ | $\mathbf{73.94}\%$ |
| PfLDS | $-505.68$ | $3.05\%$ | $6.23\%$ |

Table 2 shows the per-trial ELBO on test data for our model and for PfLDS. Again, our model has a larger ELBO than PfLDS, even when PfLDS has access to 20 times more hidden dimensions: our method is more successful at producing low-dimensional representations of trials than PfLDS. The table also shows the percentage of correctly predicted test trial types when using 15-nearest neighbors on the posterior means of train data (the entire trajectories are used for PfLDS). While $23.73\%$ might seem small, it should be noted that it is significantly better than random guessing (which would have $2.5\%$ accuracy) and that the model was not trained to minimize this objective. Regardless, we can see that our method also outperforms PfLDS in this metric, even when using a much lower-dimensional representation of each trial. The last entry in the table shows the percentage of explained variation when doing ANOVA on the test posterior means (denoted SSG/SST), using trial type as groups. Once again, we can see that our model recovers a more meaningful representation of the trials.

### 4.2.2 CYCLING DATA

We also fit our model to a not yet published dataset collected by our collaborators from the Churchland lab at Columbia University. After preprocessing (see supplementary material), it consists of 1300 and 188 train and test trials, respectively. During each trial, 20 neurons of a primate are measured as it pedals for approximately $8s$. There are 8 types of trials, based on whether the primate is pedaling forwards or backwards and at which speed.

We use the same hyperparameter settings as for the reaching data, except we use 26 uniformly spaced knots (25 intervals) and 28 bins for PfLDS, as well as a hidden dimension $m = 10$, resulting in hidden trajectories of dimension 280 for PfLDS. Table 3 quantitatively compares our method against PfLDS. The ELBO is actually higher for PfLDS, which appears (in preliminary analysis not shown) to be caused by an artifact of preprocessing the data rather than any essential performance loss.

While the ELBO was better for PfLDS, the quality of our latent representations is significantly better, as shown by the accuracy of 15-nearest neighbors to predict test trial types (random guessing would have $12.5\%$ accuracy) and the ANOVA percentage of explained variation of the test posterior means. This is particularly impressive as our latent representations have 28 times fewer dimensions. We did experiment with different hyperparameter settings, and found that the ELBO of PfLDS increased

Table 3: Quantitative comparison of our method (DRS-VAE) against PfLDS on cycling data.

| MODEL | ELBO | 15-NN | SSG/SST |
|---|---|---|---|
| DRS-VAE | 6372.5 | **55.85**% | **70.03**% |
| PfLDS | **6532**.5 | 11.70% | 3.17% |

slightly when using more time bins (at the cost of even higher-dimensional latent representations), whereas our ELBO remained the same when increasing the number of intervals. However, even in this setting the accuracy of 15-nearest neighbors and the percentage of explained variation did not improve for PfLDS.

## 5 CONCLUSIONS

In this paper we introduced Deep Random Splines, an alternative to Gaussian processes to model random functions. Owing to our key modeling choices and use of results from the spline and optimization literatures, fitting DRS is tractable and allows one to enforce shape constraints on the random functions. While we only enforced nonnegativity and smoothness in this paper, it is straightforward to enforce constraints such as monotonicity (or convexity/concavity). We also proposed a variational autoencoder that takes advantage of DRS to accurately model and produce meaningful low-dimensional representations of neural activity.

Future work includes using DRS-VAE for multi-dimensional point processes, for example spatial point processes. While splines would become harder to use in such a setting, they could be replaced by any family of easily-integrable nonnegative functions, such as, for example, conic combinations of Gaussian kernels. Another line of future work involves using a more complicated point process than the Poisson, for example a Hawkes process, by allowing the parameters of the spline in a certain interval to depend on the previous spiking history of previous intervals. Finally, DRS can be applied in more general settings than the one explored in this paper since they can be used in any setting where a random function is involved, having many potential applications beyond what we analyzed here.

### ACKNOWLEDGMENTS

We thank Sean Perkins, Karen Schroeder and Mark Churchland for sharing the cycling data. This work was supported by the following grants: Simmons Foundation 542963, NIH NINDS 5R01NS100066, the McKnight Endowment Fund, NSF 1707398 and The Gatsby Charitable Foundation (JPC).

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

# Supplementary Material

## Parametrization for nonnegative splines of even degree

As mentioned on section 2.1, there is an alternative characterization of nonnegative polynomials of even degree $d = 2k$ on an interval $[l, u)$ that is analogous to equation 1 of the manuscript, which says that the polynomial $p$ is nonnegative on the interval if and only if it can be written as:

$$p(t) = [t]^\top Q_1[t] + (u - t)(t - l)[\tilde{t}]^\top Q_2[\tilde{t}]$$

where again, $[t] = (1, t, t^2, \ldots, t^k)^\top$ and $Q_1$ is a $(k+1) \times (k+1)$ symmetric positive semidefinite matrix. In this case $Q_2$ is now a $k \times k$ symmetric positive semidefinite matrix and $[\tilde{t}] = (1, t, t^2, \ldots, t^{k-1})^\top$. Again, it follows that a piecewise polynomial of degree $d$ with knots $t_0, \ldots, t_I$ defined as $p^{(i)}(t)$ for $t \in [t_{i-1}, t_i)$ for $i = 1, \ldots, I$ is nonnegative if and only if it can be written as:

$$p^{(i)}(t) = [t]^\top Q_1^{(i)}[t] + (t_i - t)(t - t_{i-1})[\tilde{t}]^\top Q_2^{(i)}[\tilde{t}]$$

for $i = 1, \ldots, I$, where each $Q_1^{(i)}$ is a $(k+1) \times (k+1)$ symmetric positive semidefinite matrix and each $Q_2^{(i)}$ is a $k \times k$ symmetric positive semidefinite matrix.

## Projecting onto the space of smooth splines

As mentioned in section 2.2, mapping to $\Psi = \cap_{j=0}^{s+1} \mathcal{C}_j$ can be achieved through the method of alternating projections. As mentioned previously, projecting onto $\mathcal{C}_0$ can be easily done through eigen-decomposition. We now go through the details on how to project onto the other $\mathcal{C}_j$ sets. We will only cover $\mathcal{C}_1$, $\mathcal{C}_2$ and $\mathcal{C}_3$ for odd-degree splines as we used splines of degree 3 and smoothness 2, but projecting onto $\mathcal{C}_j$ for $j \geq 4$ for higher degree splines can be done in an analogous way. Projections for even degree splines can also be derived in an analogous way.

### Continuity projection for splines of odd degree

Suppose we are given $(Q_1^{(i)}, Q_2^{(i)})_{i=1}^I$, which are $(k+1) \times (k+1)$ matrices (not necessarily in $\Psi$), defining a piecewise polynomial as in equation 2 of the manuscript. Computing the projection $(X_*^{(i)}, Y_*^{(i)})_{i=1}^I$ of $(Q_1^{(i)}, Q_2^{(i)})_{i=1}^I$ onto $\mathcal{C}_1$ can be done by solving the following optimization problem:

$$(X_*^{(i)}, Y_*^{(i)})_{i=1}^I = \underset{(X^{(i)}, Y^{(i)})_{i=1}^I}{\arg\min} \sum_{i=1}^I ||X^{(i)} - Q_1^{(i)}||_F^2 + ||Y^{(i)} - Q_2^{(i)}||_F^2$$

$$\text{s.t. } (t_i - t_{i-1})[t_i]^\top Y^{(i)}[t_i] = (t_{i+1} - t_i)[t_i]^\top X^{(i+1)}[t_i], \text{ for } i = 1, \ldots, I - 1$$

where $|| \cdot ||_F$ denotes the Frobenius norm and each constraint is merely forcing the piecewise function to be continuous at knot $i$ for $i = 1, \ldots, I-1$. Note that this is a quadratic optimization problem with linear constraints, and can be solved analytically. The corresponding Lagrangian is:

$$L((X^{(i)}, Y^{(i)})_{i=1}^I, \lambda) = \sum_{i=1}^I ||X^{(i)} - Q_1^{(i)}||_F^2 + ||Y^{(i)} - Q_2^{(i)}||_F^2$$

$$+ \sum_{i=1}^{I-1} \lambda_i \Big( (t_i - t_{i-1})[t_i]^\top Y^{(i)}[t_i] - (t_{i+1} - t_i)[t_i]^\top X^{(i+1)}[t_i] \Big)$$

where $\lambda = (\lambda_1, \dots, \lambda_{I-1})^\top \in \mathbb{R}^{I-1}$. By solving the KKT conditions, it can be verified that:

$$\begin{cases} X_*^{(i)} = Q_1^{(i)} + \frac{\lambda_{i-1}^*}{2} A_{i-1} \text{ , for } i = 1, \dots, I \\ Y_*^{(i)} = Q_2^{(i)} - \frac{c_i \lambda_i^*}{2} A_i \text{ , for } i = 1, \dots, I \\ \lambda_i^* = \frac{2}{1+c_i^2} \frac{[t_i]^\top (c_i Q_2^{(i)} - Q_1^{(i+1)})[t_i]}{([t_i]^\top [t_i])^2} \text{ , for } i = 1, \dots, I - 1 \end{cases}$$

where $c_i = \frac{t_i - t_{i-1}}{t_{i+1} - t_i}$ for $i = 1, \dots, I - 1$, $c_I = 0$, $\lambda_0^* = 0$, $\lambda_I^* = 0$ and $A_i = [t_i][t_i]^\top$ for $i = 0, \dots, I$.

## Differentiability projection for splines of odd degree

Analogously, computing the projection $(X_*^{(i)}, Y_*^{(i)})_{i=1}^I$ of $(Q_1^{(i)}, Q_2^{(i)})_{i=1}^I$ onto $\mathcal{C}_2$ can be done by solving the following optimization problem:

$$(X_*^{(i)}, Y_*^{(i)})_{i=1}^I = \underset{(X^{(i)}, Y^{(i)})_{i=1}^I}{\arg\min} \sum_{i=1}^I ||X^{(i)} - Q_1^{(i)}||_F^2 + ||Y^{(i)} - Q_2^{(i)}||_F^2$$

$$\text{s.t. } -[t_i]^\top X^{(i)}[t_i] + [t_i]^\top Y^{(i)}[t_i]$$
$$+ (t_i - t_{i-1})[t_i']^\top Y^{(i)}[t_i] + (t_i - t_{i-1})[t_i]^\top Y^{(i)}[t_i']$$
$$= -[t_i]^\top X^{(i+1)}[t_i] + (t_{i+1} - t_i)[t_i']^\top X^{(i+1)}[t_i]$$
$$+ (t_{i+1} - t_i)[t_i]^\top X^{(i+1)}[t_i'] + [t_i]^\top Y^{(i+1)}[t_i], \text{ for } i = 1, \dots, I - 1$$

where $[t'] = (0, 1, 2t, 3t^2, \dots, kt^{k-1})^\top$ and each constraint is now forcing the values of the left and right derivatives of the piecewise function to match at knot $i$ for $i = 1, \dots, I - 1$. Again, this is a quadratic optimization problem with linear constraints. By writing the Lagrangian and solving the KKT conditions, we get:

$$\begin{cases} X_*^{(i)} = Q_1^{(i)} + \frac{\lambda_i^*}{2} A_i - \frac{\lambda_{i-i}^*}{2}(A_{i-1} - (t_i - t_{i-1})M_{i-1}) \text{ , for } i = 1, \dots, I \\ Y_*^{(i)} = Q_2^{(i)} - \frac{\lambda_i^*}{2}(A_i + (t_i - t_{i-1})M_i) + \frac{\lambda_{i-1}^*}{2} A_{i-1} \text{ , for } i = 1, \dots, I \end{cases}$$

where $M_i = [t_i][t_i']^\top + [t_i'][t_i]^\top$ for $i = 0, \dots, I$ and:

$$\lambda_{i-1}^* \left( [t_i]^\top (A_{i-1} - \frac{t_i - t_{i-1}}{2} M_{i-1})[t_i] + (t_i - t_{i-1})[t_i']^\top A_{i-1}[t_i] \right)$$
$$+ \lambda_i^* \left( [t_i]^\top (-2A_i - \frac{t_{i+1} - 2t_i + t_{i-1}}{2} M_i)[t_i] \right.$$
$$\left. + (t_{i+1} - 2t_i + t_{i-1} - (t_i - t_{i-1})^2 - (t_{i+1} - t_i)^2)[t_i']^\top M_i[t_i] \right)$$
$$+ \lambda_{i+1}^* \left( [t_i]^\top (A_{i+1} + \frac{t_{i+1} - t_i}{2} M_{i+1})[t_i] - (t_{i+1} - t_i)[t_i']^\top A_{i+1}[t_i] \right)$$
$$= [t_i]^\top (Q_1^{(i)} - Q_1^{(i+1)} - Q_2^{(i)} + Q_2^{(i+1)})[t_i] + 2[t_i']^\top ((t_{i+1} - t_i)Q_1^{(i+1)} - (t_i - t_{i-1})Q_2^{(i)})[t_i]$$

for $i = 1, \dots, I - 1$ and again, $\lambda_0^* = 0$ and $\lambda_I^* = 0$. This is a tridiagonal system of $I - 1$ linear equations with $I - 1$ unknowns and can be solved efficiently in $O(I)$ time with simplified Gaussian elimination.

## Second differentiability projection for splines of odd degree

Finally, computing the projection $(X_*^{(i)}, Y_*^{(i)})_{i=1}^I$ of $(Q_1^{(i)}, Q_2^{(i)})_{i=1}^I$ onto $\mathcal{C}_2$ can be done by solving the following optimization problem:

$$(X_*^{(i)}, Y_*^{(i)})_{i=1}^I = \underset{(X^{(i)}, Y^{(i)})_{i=1}^I}{\arg\min} \sum_{i=1}^I ||X^{(i)} - Q_1^{(i)}||_F^2 + ||Y^{(i)} - Q_2^{(i)}||_F^2$$

$$\text{s.t. } -2[t_i']^\top X^{(i)}[t_i] - 2[t_i]^\top X^{(i)}[t_i'] + 2[t_i']^\top Y^{(i)}[t_i] + 2[t_i]^\top Y^{(i)}[t_i']$$
$$+ (t_i - t_{i-1})[t_i'']^\top Y^{(i)}[t_i] + 2(t_i - t_{i-1})[t_i']^\top Y^{(i)}[t_i'] + (t_i - t_{i-1})[t_i]^\top Y^{(i)}[t_i'']$$
$$= -2[t_i']^\top X^{(i+1)}[t_i] - 2[t_i]^\top X^{(i+1)}[t_i'] + (t_{i+1} - t_i)[t_i'']^\top X^{(i+1)}[t_i]$$
$$+ 2(t_{i+1} - t_i)[t_i']^\top X^{(i+1)}[t_i'] + (t_{i+1} - t_i)[t_i]^\top X^{(i+1)}[t_i''] + 2[t_i']^\top Y^{(i+1)}[t_i]$$
$$+ 2[t_i]^\top Y^{(i+1)}[t_i']$$

where $[t''] = (0, 0, 2, 6t, \ldots, k(k-1)t^{k-2})^\top$ and each constraint is now forcing the values of the left and right second derivatives of the piecewise function to match at knot $i$ for $i = 1, \ldots, I-1$. Again, this is a quadratic optimization problem with linear constraints. By writing the Lagrangian and solving the KKT conditions, we get:

$$\begin{cases} X_*^{(i)} = Q_1^{(i)} + \lambda_i^* M_i - \frac{\lambda_{i-i}^*}{2} B_{i-1} \text{ , for } i = 1, \ldots, I \\ Y_*^{(i)} = Q_2^{(i)} - \frac{\lambda_i^*}{2} E_i + \lambda_{i-1}^* M_{i-1} \text{ , for } i = 1, \ldots, I \end{cases}$$

where $B_{i-1} = 2M_{i-1} - (t_i - t_{i-1})([t''_{i-1}][t_{i-1}]^\top + 2[t'_{i-1}][t'_{i-1}]^\top + [t_{i-1}][t''_{i-1}]^\top)$ and $E_i = 2M_i - (t_i - t_{i-1})([t''_i][t_i]^\top + 2[t'_i][t'_i]^\top + [t_i][t''_i]^\top)$ for $i = 1, \ldots, I$ and:

$$\lambda_{i-1}^* \Big([t'_i]^\top (2B_{i-1} + 4M_{i-1})[t_i] + 2(t_i - t_{i-1})[t''_i]^\top M_{i-1}[t_i] + 2(t_i - t_{i-1})[t'_i]^\top M_{i-1}[t'_i]\Big)$$

$$+ \lambda_i^* \Big([t'_i]^\top (-8M_i - 2E_i - 2B_i)[t_i] + [t''_i]^\top ((t_{i+1} - t_i)B_i - (t_i - t_{i-1})E_i)[t_i]$$

$$+ [t'_i]^\top ((t_{i+1} - t_i)B_i - (t_i - t_{i-1})E_i)[t'_i]\Big)$$

$$+ \lambda_{i+1}^* \Big([t'_i]^\top (E_{i+1} + 4M_{i+1})[t_i] - 2(t_{i+1} - t_i)[t''_i]^\top M_{i+1}[t_i] - 2(t_{i+1} - t_i)[t'_i]^\top M_{i+1}[t'_i]\Big)$$

$$= 4[t'_i]^\top (Q_1^{(i)} - Q_1^{(i+1)} - Q_2^{(i)} + Q_2^{(i+1)})[t_i] + 2[t''_i]^\top ((t_{i+1} - t_i)Q_1^{(i+1)} - (t_i - t_{i-1})Q_2^{(i)})[t_i]$$

$$+ 2[t'_i]^\top ((t_{i+1} - t_i)Q_1^{(i+1)} - (t_i - t_{i-1})Q_2^{(i)})[t'_i] \text{ , for } i = 1, \ldots, I-1$$

where again, $\lambda_0^* = 0$ and $\lambda_I^* = 0$. Again, this is a tridiagonal system of $I-1$ linear equations with $I-1$ unknowns that can be solved efficiently.

## Reaching Data Preprocessing

We include only successful trials (i.e. when the primate reaches to the correct location) and use only spikes occurring in a window of $-100ms$ and $300ms$ from the time that movement starts. We also reduce the total number of neurons as inference with our method requires one LSTM per neuron and having too many neurons renders training slow. In order to do so, we use the following GLM:

$$y_r \sim \text{Multinomial}\big(C, \text{softmax}(\tilde{K}_{r,\cdot}^\top \beta)\big)$$

where $y_r$ is the trial type of trial $r$, $C = 40$ is the number of trial types, $\tilde{K}_{r,\cdot} \in \mathbb{R}^N$ is a vector containing the (centered and standardized) number of spikes in trial $r$ for each of the $N = 223$ neurons, and $\beta \in \mathbb{R}^{N \times C}$ are the GLM parameters. We train the GLM using group lasso (Yuan and Lin, 2006), where the groups are defined by neurons. That is, the GLM is trained through maximum likelihood with an added penalty:

$$\lambda \sum_{n=1}^N ||\beta_{n,\cdot}||_2^2$$

where $\beta_{n,\cdot}$ is the $n^{th}$ row of $\beta$. This makes it so that the coefficients in each group hit zero simultaneously. A neuron $n$ is removed if $||\hat{\beta}_{n,\cdot}|| = 0$. We use a regularization parameter $\lambda$ such that all but 20 neurons are removed. This provides a principled way of reducing the number of neurons while making sure that the kept neurons are useful. As PfLDS does not require the use of LSTMs, it can be run on the data without removing neurons. While doing this did increase performance of PfLDS, it did so very marginally and our model still heavily outperformed PfLDS.

## Cycling Data Preprocessing

Once again, we only keep successful trials (i.e. when the primate pedals in the correct direction and speed) and reduce the total number of neurons $N = 256$ to 20 by using group lasso. Since each trial has a different length, we extend every trial to have the same length as the longest trial. We add no spikes to these extended time periods. We also tried running PfLDS with all the neurons and saw only a very marginal improvement, like we did with reaching data.

