# OpenReview forum: "Deep Random Splines for Point Process Intensity Estimation"
_ICLR.cc/2019/Workshop/DeepGenStruct — DeepGenStruct 2019_

### Official Review · AnonReviewer2 · 2019-04-10
**Deep Random Splines, a class of random functions that can incorporate constraints such as non-negativity, monotonicity, or convexity.**

**Rating:** 5
**Confidence:** 2

**Review:**

This paper presents the DRS, a class of random functions. In contrast to GPs, DRS uses splines to define the function; in particular it defines a piecewise function using a spline on each interval, making sure that the overall function is continuous (and so are its derivatives). The parameters of each spline are modeled using a deep neural network that takes Gaussian noise as input. Importantly, the parameters of each spline can be chosen so that the modeled function satisfies certain desired shape properties, such as non-negativity, monotonicity, or convexity. As a use case, the paper shows an example of application that uses non-negative splines to model the rate of a Poisson process, analogous to the log-Gaussian Cox process. Inference is carried out with amortized variational inference.

I found this is a strong paper and therefore recommend acceptance. It is well explained and the method is of interest to the community.

One question that I had is how this model performs/scales with the dimensionality of the output space (i.e., the analogous to the multi-output GP).

Another interesting point would be to compare against the log-Gaussian Cox process in the experiments.

As a minor comment, from page 3 I didn't understand why the Q matrices are of size 2x2 when d=3. On page 2, it says that each matrix Q if of size k-by-k when d=2k+1, so for d=3 shouldn't we obtain k=1?

---

### Official Review · AnonReviewer1 · 2019-04-17
**A combination of a neural network with input random noise  to model one-dimensional random functions using splines**

**Rating:** 3
**Confidence:** 2

**Review:**

The paper presents a method to model univariate functions in the interval [T1, T2] that makes use of splines having random parameters modelled by a neural network. The overall method is suitable only for modelling univariate functions (e.g. when the input is time) and the authors apply this model to Poisson point processes. A nice feature of the approach is  that the spline-based formulation allows to enforce non-negativity constraints to the random functions. Thus, such random functions can be used to model the intensity of Poisson processes without needing to impose a non-negativity transformation.

I think that the claim in the introduction that the proposed Deep Random Splines is an alternative to Gaussian processes
is an overstatement. This is because the current method is only suitable for univariate functions and even trying to extend it to two-dimensional input spaces is challenging. I would suggest the authors to extend their method to two and three-dimensional domains so that their algorithm  can have a bit broader applicability in some spatio-temporal problems.

---

### Decision · Program_Chairs · 2019-04-19
**Acceptance Decision**

Accept